# DeSIQ: Towards an Unbiased, Challenging Benchmark for Social Intelligence Understanding

Xiao-Yu Guo , Yuan-Fang Li , and Gholamreza Haffari

Faculty of Information Technology, Monash University, Melbourne, Australia
{xiaoyu.guo,yuanfang.li,gholamreza.haffari}@monash.edu

## Abstract

Social intelligence is essential for understanding and reasoning about human expressions, intents and interactions. One representative benchmark for its study is Social Intelligence Queries (Social-IQ), a dataset of multiple-choice questions on videos of complex social interactions. We define a comprehensive methodology to study the soundness of Social-IQ, as the soundness of such benchmark datasets is crucial to the investigation of the underlying research problem. Our analysis reveals that Social-IQ contains substantial biases, which can be exploited by a moderately strong language model to learn spurious correlations to achieve perfect performance without being given the context or even the question. We introduce DeSIQ, a new challenging dataset, constructed by applying simple perturbations to Social-IQ. Our empirical analysis shows DeSIQ significantly reduces the biases in the original Social-IQ dataset. Furthermore, we examine and shed light on the effect of model size, model style, learning settings, commonsense knowledge, and multi-modality on the new benchmark performance. Our new dataset, observations and findings open up important research questions for the study of social intelligence.

## 1 Introduction

Social intelligence is a long-standing research area in social science and psychology (Thorndike, 1920; Andreou, 2006; Goleman, 2007). It is the capacity to understand and navigate complex social situations. Social intelligence is more than the perception of objects and human actions, as it requires a deeper understanding of human intents and interactions behind these actions or words.

The study of social intelligence is an emerging area in both the NLP and computer vision communities. One representative work, Social-IQ (Social Intelligence Queries) (Zadeh et al., 2019), is a benchmark dataset measuring social intelligence of current AI systems. It is a multiple choice question answering dataset with multi-modal inputs, including questions, answer options, videos, etc; see an example in Figure 1. Although Social-IQ contains rigorously human-annotated data, surprisingly, we find even small models like T5-small (Raffel et al., 2020) could easily achieve 100% answer option accuracy (Table 3).

The perfect performance of such an underpowered model prompted us to conduct further investigation to identify its source. Through employing different models and perturbation methods on the answer options, we identify significant biases in the Social-IQ dataset, in which the representations of correct and incorrect options are easily separable, regardless of the questions (Figure 3). Thus, the models are able to exploit such a *shortcut* (Jia and Liang, 2017; Jiang and Bansal, 2019) to answer questions with a high accuracy, without necessarily understanding social intelligence.

To debias the Social-IQ dataset, we propose a simple yet effective debiasing approach and present a new unbiased benchmark DeSIQ, by substituting all the incorrect answer options with correct answer options from randomly selected other questions.

We establish a performance baseline on DeSIQ with T5-small and Delphi (Jiang et al., 2021), a language model pretrained with commonsense and social norms knowledge. Given answer options only or question-answers, both T5-small and Delphi obtain close to random accuracy. By making use of multi-modal inputs, both T5-small and Delphi achieve an accuracy of up to 77%. These results demonstrate that DeSIQ is unbiased and challenging. Interestingly, both models also outperform GPT-3 and ChatGPT, further indicating the challenging nature of the social intelligence understanding problem.

Our contributions are:

- We propose six formally defined methods to identify the bias in Social-IQ. From the answer pertur-

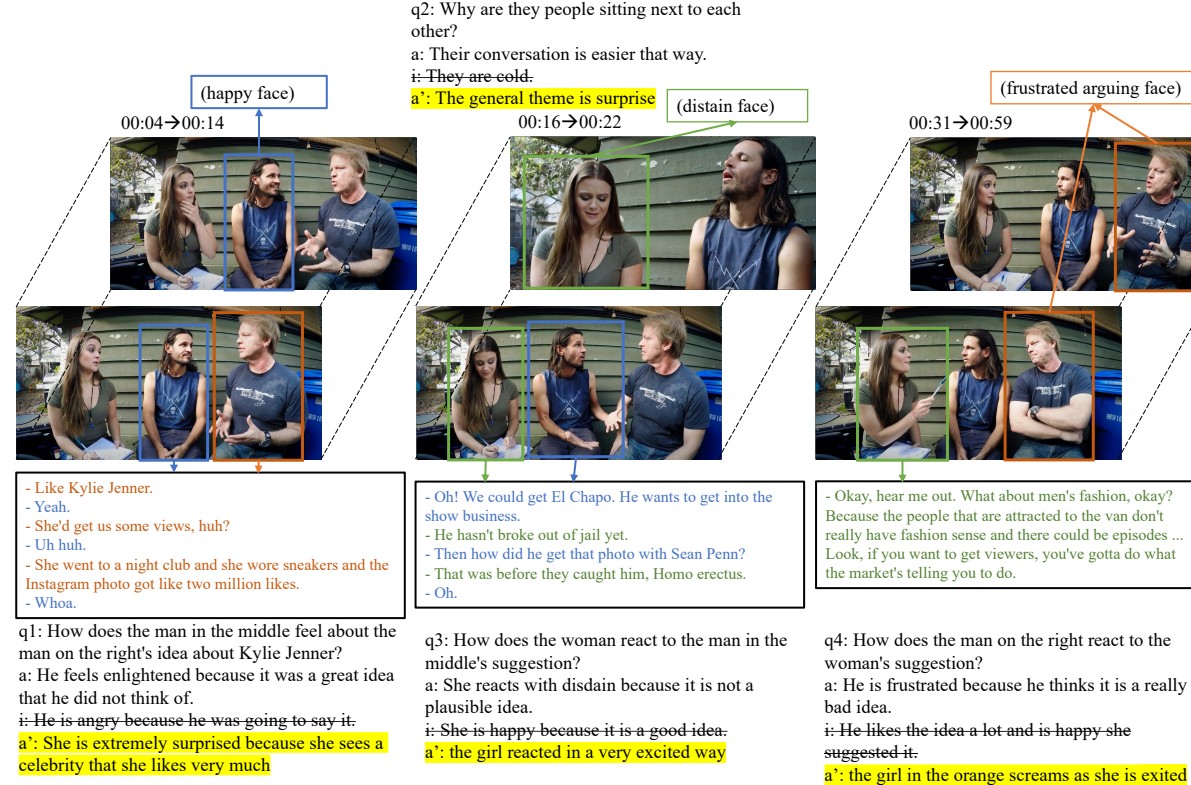

q2: Why are they people sitting next to each other?
a: Their conversation is easier that way.
~~i: They are cold.~~
a': The general theme is surprise

(happy face)

00:04→00:14

(distain face)

00:16→00:22

(frustrated arguing face)

00:31→00:59

- Like Kylie Jenner.
- Yeah.
- She'd get us some views, huh?
- Uh huh.
- She went to a night club and she wore sneakers and the Instagram photo got like two million likes.
- Whoa.

- Oh! We could get El Chapo. He wants to get into the show business.
- He hasn't broke out of jail yet.
- Then how did he get that photo with Sean Penn?
- That was before they caught him, Homo erectus.
- Oh.

- Okay, hear me out. What about men's fashion, okay? Because the people that are attracted to the van don't really have fashion sense and there could be episodes ... Look, if you want to get viewers, you've gotta do what the market's telling you to do.

q1: How does the man in the middle feel about the man on the right's idea about Kylie Jenner?
a: He feels enlightened because it was a great idea that he did not think of.
~~i: He is angry because he was going to say it.~~
a': She is extremely surprised because she sees a celebrity that she likes very much

q3: How does the woman react to the man in the middle's suggestion?
a: She reacts with disdain because it is not a plausible idea.
~~i: She is happy because it is a good idea.~~
a': the girl reacted in a very excited way

q4: How does the man on the right react to the woman's suggestion?
a: He is frustrated because he thinks it is a really bad idea.
~~i: He likes the idea a lot and is happy she suggested it.~~
a': the girl in the orange screams as she is exited

Figure 1: One example in the Social-IQ and DeSIQ benchmark. For Social-IQ, q, a, i stand for question, correct and incorrect answer respectively, while a' with yellow background color is the unbiased incorrect answer we substitute in DeSIQ. Different colors represent different persons, including the facial expressions and oral speaking words. The transcripts are in three black squares related to certain video clips in the above.

bation experiments, we find that the bias mainly exists in the answer options.

- We propose DeSIQ, an unbiased, and more challenging multi-modal question answering benchmark, designed to better measure social intelligence for machine learning models.

- We propose two effective models that outperform the baseline and GPT-3/ChatGPT on our new benchmark. We also make detailed analysis and comparison on the performance of these models.

## 2 Identifying Biases in Social-IQ

### 2.1 The Social Intelligence Datasets

Social-IQ (Zadeh et al., 2019) is an unconstrained multi-modal, multiple-choice question answering (MCQA) dataset designed to evaluate the social intelligence of machine learning models. It contains videos about social interactions, questions and multiple-choice answer options, in which the questions and answer options were crowdsourced. For each video, the context for all questions and answer options includes not only the original video, but also the corresponding extracted audio and au-

| Number | Training | Development | Total |
|---|---|---|---|
| Video | 888 | 127 | 1,015 |
| Question | 5,328 | 762 | 6,090 |
| Correct | 21,312 | 3,048 | 24,360 |
| Incorrect | 15,984 | 2,286 | 18,270 |

Table 1: Statistics of the Social-IQ dataset. On average, each video has 6 questions; for each question, there are 4 correct answer options and 3 incorrect answer options.

| Number | Training | Development | Total |
|---|---|---|---|
| Video | 987 | 145 | 1,132 |
| Question | 6,159 | 943 | 7,102 |
| Correct | 6,159 | 943 | 7,102 |
| Incorrect | 18,477 | 2,829 | 21,306 |

Table 2: Statistics of the Social-IQ-2.0 dataset. For each question, there is only 1 correct answer option.

tomatically generated transcripts[1]. Detailed dataset statistics are shown in Table 1.

Social-IQ provides two configurations: A2 (2-way, i.e. one correct answer option and one incorrect option for each question) and A4 (2-way, i.e. one correct option and 3 incorrect options for each question) for training and evaluation, in which

---

[1] We don't have access to the raw transcript, video and audio data so we use extracted features downloaded from https://github.com/matsuolab/CMU-MultimodalSDK.

model performance is measured using binary and 4-way accuracy respectively.

Most recently, Social-IQ-2.0 was released online[2] with the A4 configuration. Though nearly half of the videos overlap with Social-IQ, almost all questions and answers were newly annotated. Moreover, raw video and audio files have been provided instead of only features in the original Social-IQ dataset. The detailed statistics are shown in Table 2. For simplicity, **v1** and **v2** represent Social-IQ and Social-IQ-2.0 respectively, which are used interchangeably.

## 2.2 Methodology

In this section, we propose several experimental settings to identify biases in a MCQA dataset. Let $q$ and $q'$ denote two different questions, $a$ and $i$ denote the correct and an incorrect answer option of $q$ respectively, and $a'$ and $i'$ denote the correct and an incorrect answer option of $q'$ respectively. We define six methods to identify biases:

**No context and question (NCAQ):** the contexts and questions for all answer options are removed. I.e., the model is only given all answer options.

An MCQA dataset should be sufficiently challenging that no model can predict a correct answer when neither the input context nor the question is not provided.

**More Powerful Model (MPM):** the model is substituted by a larger, more capable model.

It is plausible to induce a performance increase on the dataset when a stronger model (e.g. with more trainable parameters and/or one that is fine-tuned on relevant data) is employed. However, a sufficiently hard dataset should not induce a perfect model performance (i.e. near 100% accuracy score). This can be tested with models of different sizes and thus capabilities.

**RIWI:** Replace $i$ with $i'$, $(q, a, i) \rightarrow (q, a, i')$

**RIWA:** Replace $i$ with $a'$, $(q, a, i) \rightarrow (q, a, a')$

**RAWI:** Replace $a$ with $i'$, $(q, a, i) \rightarrow (q, i', i)$

**RAWA:** Replace $a$ with $a'$, $(q, a, i) \rightarrow (q, a', i)$

With the above perturbations, we expect the dataset to induce the following robustness behaviours. With **RIWI** or **RIWA** applied to the dev/test set, we should expect that a model's performance should not significantly deviate from

[2] https://cmu-multicomp-lab.github.io/social-iq-2.0/

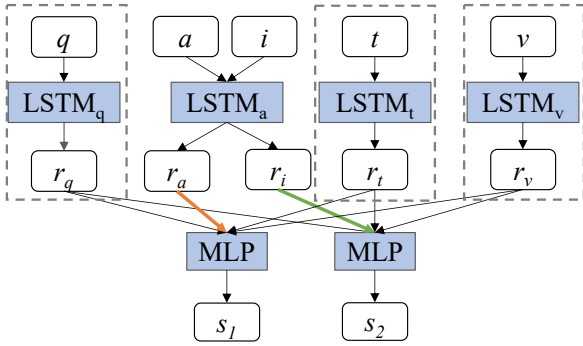

Figure 2: Overall architecture of the LSTM baseline (Zadeh et al., 2019). $q, a, i, t, v$ denote question, correct answer, incorrect answer, transcript and video features. $r_q, r_a, r_i, r_t, r_v$ are corresponding features extracted using different LSTMs. Dashed squares represent optional input features. Two multi-layer perception (MLP) are parameter-shared. The output will be two scores $s_1, s_2$ respectively of the correct and incorrect answer options.

the original dataset. With **RAWI** or **RAWA**, the model should perform significantly worse.

## 2.3 Biases in Social-IQ

We evaluate the A2 (binary choice) configuration and A4 (multiple choice) configurations of Social-IQ, and A4 configuration of Social-IQ-2.0 in the experimental settings discussed above, and surprisingly, we observe that they are both biased. Below, we describe our detailed analysis and show that Social-IQ contains substantial biases that can be exploited by moderately strong language models. Table 3 summarises the experimental results. In the fully supervised setting, we evaluate the performance of the LSTM-based model in the original Social-IQ paper (Zadeh et al., 2019) (Figure 2) as well as the more capable T5-small (Raffel et al., 2020), which we use as the encoder to replace the LSTM in Figure 2.

**Evidence of Dataset Bias.** We start from the **NCAQ** settings, i.e., only the answer options ($a$ and $i$) are given as model input, without the question and video, for both training and evaluation. Under this setting, we also compare models' performance with different perturbations on the answer options. Table 3 shows that the basic LSTM model outperforms the random guess by 9.45% on **v1** (i.e. Social-IQ). With the unreasonable inputs (with no context nor question), these accuracy scores show that the Social-IQ dataset is biased.

We postulate that while a stronger model (i.e. **MPM**) should obtain better performance than LSTM, without being given sufficient information,

| Data | Model | Settings | A2 | A4 |
|------|-------|----------|-----|-----|
|  | Random | none | 50 | 25 |
|  | LSTM | NCAQ | 59.45 | 34.84 |
| **v1** | T5-small (MPM) | NCAQ | **100** | **100** |
|  |  | NCAQ+RIWI | 97.37 | 99.97 |
|  |  | NCAQ+RIWA | 50.21 | 25.03 |
|  |  | NCAQ+RAWI | 49.93 | 23.76 |
|  |  | NCAQ+RAWA | 97.25 | **100** |
| **v2** | T5-small (MPM) | NCAQ | - | **63.35** |
|  |  | NCAQ+RIWI | - | 59.66 |
|  |  | NCAQ+RIWA | - | 24.72 |
|  |  | NCAQ+RAWI | - | 23.72 |
|  |  | NCAQ+RAWA | - | **62.36** |

Table 3: Model performance on A2 (binary choice) and A4 (multiple choice) under different experimental settings, in which only answer options are given as model inputs (but not questions nor context). '-' represents the results are inapplicable.

even the stronger model should not perform unreasonably well. Thus, we experiment with T5-small, a modestly-sized yet more capable model. As it can be seen in Table 3, T5-small outperforms LSTM by a large margin on **v1**. Surprisingly, it also achieves a perfect 100% accuracy score on **v1** and 63.35% on **v2** without being given the context nor the question. These results provide strong evidence of the biases in these datasets.

Finally, we study the other four perturbation settings by applying them to the dev sets. Below we analyse the performance on **v1** in detail, followed by a discussion on **v2**.

- **RIWI.** Similar to the performance on the original dataset, T5-small achieves an unreasonable performance of 97.37% on A2 and 99.97% in A4. It indicates that the model can easily distinguish the correct answer from the incorrect options.

- **RIWA.** It leads to a large performance degradation: A2 from 100% to 50.21%, A4 from 100% to 25.03%, similar to random guess (i.e., 50% and 25%). This shows that T5-small is unable to distinguish the correct answer options, regardless of the question it is used for.

- **RAWI.** This produces a dataset containing only incorrect answer options. We consider the incorrect answer option that replaces the correct answer option as the correct answer.

  Intuitively, it should lead a model to randomly guess, as none of the options is correct. In Table 3, we can observe that **RAWI** leads to a large performance drop: A2 from 100% to 49.93%, A4 from 100% to 23.76%, indicating that T5-small cannot distinguish incorrect answers from each other, confirming our intuition.

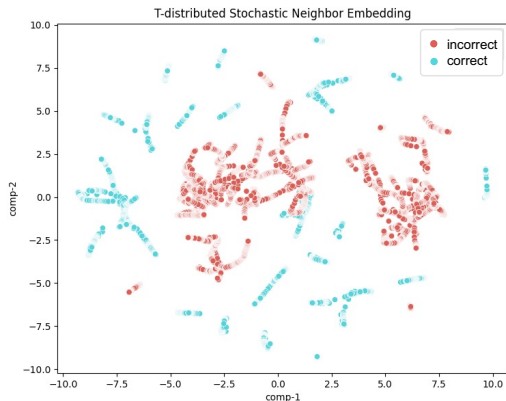

Figure 3: T-SNE visualization of correct and incorrect answer options. Red dots are incorrect answer options and blue dots are correct answer options.

- **RAWA.** It should lead to A2 with 50% and A4 with 25% performance since the correct answer option is replaced with an irrelevant correct answer of another question.

  Contrary to our intuition, **RAWA** leads to a near-perfect performance of 97.25% on A2 and even better 100% on A4.

  These unexpectedly high scores indicate that the model can easily distinguish the correct answer options from the incorrect ones of the original dataset, regardless of the question they are used for, consistent with the results of **RIWI**.

Figure 3 shows the T-distributed Stochastic Neighbor Embedding (T-SNE) visualization of the embeddings of all answer options in the Social-IQ dev set. We can observe a clear boundary between correct and incorrect answer options. The above results provide compelling evidence of the unwanted bias in Social-IQ, manifested in T5-small's strong capability in distinguishing the correct and incorrect answer options.

Similar evidence can be found in Social-IQ-2.0, as can be seen in the **v2** rows in Table 3.

## 3 DeSIQ: Debiased Social-IQ

In this section, we first describe our approach to debias Social-IQ . We then study the effectiveness of our debiasing approach and the resultant DeSIQ datasets, by comparing the performance of both LSTM and T5-small on DeSIQ in different settings.

### 3.1 Constructing DeSIQ

We propose the following perturbation-based approach to debias Social-IQ and construct a more

meaningful and challenging dataset on social intelligence. Specifically, we apply the **RIWA** perturbation on both the training and development sets of Social-IQ, ie substituting the incorrect answer options with correct answer options from the other questions. We construct two debiased datasets[3]:

- **DeSIQ$_d$.** Given an original triplet $(q, a, i)$, we randomly sample another triplet $(q', a', i')$ from *another video*. Thus, for each original triplet $(q, a, i)$, we form a new triplet $(q, a, a')$.

- **DeSIQ$_s$.** We sample $(q', a', i')$ from the *same video* for each $(q, a, i)$. Similarly, we replace the incorrect answer option $i$ with $a'$. Since $q$ and $q'$ are from the same video, their answers can have a higher chance of referring to the same entity that appears in the video. Thus, **DeSIQ$_s$** is a more challenging dataset of $(q, a, a')$.

An example video and some associated questions and answer options for both Social-IQ and DeSIQ$_s$ can be seen in Figure 1. For Social-IQ-2.0, we do the same approach to obtain **DeSIQ$_d$-2.0**.

### 3.2 Effectiveness of the Debiasing Approach

We set up a number of models in both fully supervised and zero/few-shot learning settings to show the effectiveness of our debiasing approach,

**Supervised Learning.** We train the LSTM and T5-small on Social-IQ, DeSIQ$_d$ and DeSIQ$_s$ in the same architecture (Figure 2), and train T5-small on Social-IQ-2.0. Table 4, Table 5 and Table 6 show the results, where the relevant results are shaded in gray . The second column "Input" represents the input used in both the training and evaluation procedures, where "a", "q", "t", and "v" represent answer options, the question, the transcript and the video, respectively. The third column "Concat" represents different model architectures. The symbol "✗" denotes that all inputs are separately encoded as in Figure 2, which is the focus of this subsection. The symbol "✓" denotes that all inputs are concatenated and encoded as one sequence as in Figure 4, which will be discussed in the next section.

As seen in Table 4, DeSIQ$_d$ largely reduces the bias in Social-IQ, effectively reducing the performance of both LSTM and T5-small close to random guess. For LSTM, when given only answer options, we observe a performance drop of $59.45\% \rightarrow 48.52\%$ on A2 and $34.84\% \rightarrow 27.23\%$ on A4. For

[3]Below we describe the dataset construction process for the A2 configuration. Similar perturbations are applied to the A4 configuration of Social-IQ.

---

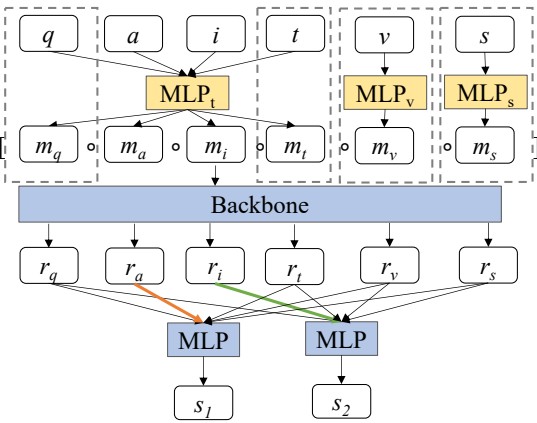

Figure 4: Our model structure to address the new benchmark DeSIQ. The A2 configuration is shown here for illustration purposes. $q, a, i, t, v, s$ are question, correct answer, incorrect answer, transcript, video and audio features; $m_q, m_a, m_i, m_t, m_v, m_s$ are the corresponding features after projection to the same dimension; and $r_q, r_a, r_i, r_t, r_v, r_s$ are the corresponding embeddings after going through the same backbone model. $[\circ]$ denotes the concatenation operation and dashed boxes denote optional input features. Three multi-layer perceptrons (MLP$_t$, MLP$_v$ and MLP$_s$) in yellow above are projectors mapping textual, video and audio features to the same space. The output will be the two scores $s_1, s_2$ (four scores for multiple-choice), representing the correct and incorrect answer option respectively.

T5-small, it suffers a larger performance drop on both A2 and A4 of DeSIQ$_d$ (to 50.16% and 34.15% respectively), although 100% A2 on Social-IQ.

The results in Table 5 show that DeSIQ$_s$ effectively reduces the bias in Social-IQ. For LSTM, DeSIQ$_s$ leads to a performance drop of $59.45\% \rightarrow 48.24\%$ on A2 and $34.84\% \rightarrow 27.06\%$. For T5-small, we can observe a performance drop of $100\% \rightarrow 48.73\%$ on A2 and of $100\% \rightarrow 33.53\%$ on A4.

Comparing results in Tables 4 and 5, we see DeSIQ$_s$ is generally more challenging than DeSIQ$_d$, i.e. compare T5-small's performance under the same settings across the two datasets. For instance, with the question feature added ("q+a" as inputs), T5-small achieves on 60.55% on DeSIQ$_d$ and 49.17% on DeSIQ$_s$ in A2, and 27.57% on DeSIQ$_d$ and 24.22% on DeSIQ$_s$ in A4.

In Table 6, DeSIQ$_d$-2.0 also reduces the bias in Social-IQ-2.0, effectively reducing the performance of T5-small close to random guess. For T5-small, it suffers a larger performance drop on A4 of DeSIQ$_d$-2.0 (63.35% to 28.07%).

**Zero-shot and Few-shot Learning.** We employ the strong GPT-3 model (Brown et al., 2020) and ChatGPT (OpenAI, 2022) with both zero-shot and

| Dataset | Input | Concat | A2 | | | A4 | | |
|---|---|---|---|---|---|---|---|---|
| | | | LSTM | T5-small | T5-small$_{Delphi}$ | LSTM | T5-small | T5-small$_{Delphi}$ |
| Social-IQ | a | ✗ | 59.45 | 100 | 100 | 34.84 | 100 | 100 |
| | q+a | ✗ | 59.78 | 100 | 100 | 38.55 | 100 | 100 |
| | q+a+t | ✗ | 60.00 | 100 | 100 | 43.84 | 100 | 100 |
| | q+a+v | ✗ | 64.38 | 100 | 100 | 46.08 | 100 | 100 |
| DeSIQ$_d$ | a | ✗ | 48.52 | 50.16 | 50.33 | 27.23 | 34.15 | 28.97 |
| | q+a | ✗ | 58.58 | 60.55 | 50.19 | 26.05 | 27.57 | 25.78 |
| | q+a+t | ✗ | 60.46 | 50.16 | 50.40 | 27.59 | 28.84 | 27.15 |
| | q+a+v | ✗ | 61.05 | 50.59 | 50.70 | 25.91 | 27.60 | 25.55 |
| DeSIQ$_d$ | a | ✓ | 49.20 | 49.52 | 50.33 | 34.85 | 36.30 | 38.18 |
| | q+a | ✓ | 61.41 | 73.47 | 75.69 | 34.91 | 62.43 | 72.91 |
| | q+a+t | ✓ | 13.17 | 74.69 | **76.77** | 29.22 | 70.80 | **74.51** |
| | q+a+v | ✓ | 56.67 | 76.72 | 74.99 | 41.77 | 72.69 | 73.24 |

Table 4: Accuracy on the Social-IQ and DeSIQ$_d$ development sets. Results shaded in gray are relevant to Sec. 3.

| Dataset | Input | Concat | A2 | | | A4 | | |
|---|---|---|---|---|---|---|---|---|
| | | | LSTM | T5-small | T5-small$_{Delphi}$ | LSTM | T5-small | T5-small$_{Delphi}$ |
| Social-IQ | a | ✗ | 59.45 | 100 | 100 | 34.84 | 100 | 100 |
| | q+a | ✗ | 59.78 | 100 | 100 | 38.55 | 100 | 100 |
| | q+a+t | ✗ | 60 | 100 | 100 | 43.84 | 100 | 100 |
| | q+a+v | ✗ | 64.38 | 100 | 100 | 46.08 | 100 | 100 |
| DeSIQ$_s$ | a | ✗ | 48.24 | 48.73 | 48.96 | 27.06 | 33.53 | 28.12 |
| | q+a | ✗ | 59.97 | 49.17 | 59.59 | 26.16 | 24.22 | 25.20 |
| | q+a+t | ✗ | 60.02 | 58.89 | 56.83 | 27.31 | 26.79 | 23.83 |
| | q+a+v | ✗ | 61.00 | 59.19 | 56.42 | 26.99 | 25.00 | 24.22 |
| DeSIQ$_s$ | a | ✓ | 48.24 | 48.73 | 48.71 | 29.67 | 29.17 | 32.32 |
| | q+a | ✓ | 59.35 | 63.08 | 63.42 | 34.73 | 62.47 | 60.58 |
| | q+a+t | ✓ | 11.52 | 65.41 | **67.70** | 22.33 | **65.23** | 51.69 |
| | q+a+v | ✓ | 51.04 | 65.96 | 65.02 | 32.56 | 56.61 | 55.05 |

Table 5: Accuracy on the Social-IQ and DeSIQ$_s$ development sets. Results shaded in gray are relevant to Sec. 3.

| Dataset | Input | Concat | A4 |
|---|---|---|---|
| Social-IQ-2.0 | a | ✗ | 63.35 |
| | q+a | ✗ | 64.63 |
| | q+a+t | ✗ | 64.06 |
| | q+a+v | ✗ | 62.28 |
| DeSIQ$_d$-2.0 | a | ✗ | 28.07 |
| | q+a | ✗ | 28.45 |
| | q+a+t | ✗ | 22.17 |
| | q+a+v | ✗ | 24.13 |
| | q+a+s | ✗ | 25.87 |
| DeSIQ$_d$-2.0 | a | ✓ | 28.07 |
| | q+a | ✓ | 57.23 |
| | q+a+t | ✓ | 52.02 |
| | q+a+v | ✓ | 68.93 |
| | q+a+s | ✓ | **74.13** |
| | q+a+t+v+s | ✓ | 37.72 |

Table 6: Accuracy on the Social-IQ-2.0 and DeSIQ$_d$-2.0 development sets. Results shaded in gray are relevant to Sec. 3.

few-shot learning to show the strength of our de-biased datasets. Social-IQ experiments are performed in the A2 configuration using GPT-3, while Social-IQ-2.0 experiments in the A4 using Chat-GPT. For zero-shot evaluation, we concatenate the question with correct and incorrect answer options (i.e. "q+a") to form the prompt[4], where the order of

the two answer options is randomly shuffled. The **zero-shot prompt** is constructed as follows:

[4]We observe that when only given answer options but not the question, GPT-3 tends to select the first option.

> "Choose the correct answer option corresponding to the question: "
> $+ q +$ " A: " $+ a +$ " B: " $+ i$

For the few-shot evaluation, we use the question similarity to find exemplars for in-context learning. For each question in the development set, we choose the top-3 most similar questions from the training set. We measure the semantic distance between questions based on their embeddings obtained from Sentence-Transformers (Reimers and Gurevych, 2019). The few-shot prompts leverage the same format as in the zero-shot evaluation, with the correct option appended to each exemplar:

> "Choose the correct answer option corresponding to the question: "
> $+ (q' +$ " A: " $+ a' +$ " B: " $+ i' +$ "A or B")*3 + **zero-shot prompt**

Table 7 shows the results. For Social-IQ, under the zero-shot setting, GPT-3 can obtain 58.26% with "q+a" and 64.63% with "q+a+t" on Social-IQ. In comparison, under either zero-shot or few-shot setting, both the DeSIQ$_d$ and DeSIQ$_s$ dataset lead

| Dataset | Input | A2 (GPT-3)/A4 (ChatGPT) | |
|---|---|---|---|
| | | Zero-shot | Few-shot |
| Social-IQ | q+a | 58.26 | 56.22 |
| | q+a+t | 64.63 | - |
| $DeSIQ_d$ | q+a | 54.78 | 54.13 |
| | q+a+t | 59.79 | - |
| $DeSIQ_s$ | q+a | 54.39 | 53.29 |
| | q+a+t | 60.13 | - |
| $DeSIQ_d$-2.0 | q+a | 59.61 | 58.02 |
| | q+a+t | 59.24 | - |

Table 7: GPT-3 performance on the A2 of Social-IQ and DeSIQ, ChatGPT performance on the A4 of DeSIQ-2.0.

to a performance drop of more than $4\%$. Under the few-shot setting for "q+a", GPT-3 does not seem to learn *shortcuts*, as the performance is unchanged compared to the zero-shot setting[5]. These results show that $DeSIQ_d$ and $DeSIQ_s$ are less biased and more challenging than Social-IQ. For Social-IQ-2.0, the performance does not change that much when leveraging ChatGPT under both zero-shot and few-shot learning settings, which also proves it is less biased than Social-IQ.

## 4 Setting Baseline Performance on DeSIQ

For our more challenging DeSIQ benchmark, we introduce a new baseline model to better handle multi-modal inputs. Its architecture is shown in Figure 4. Compared with the model in Figure 2, we add three more projection layers (three yellow MLPs) to map the original feature representations into the same dimensions. We then concatenate all the resulting representations as the inputs to a backbone MPM. For $DeSIQ_d$-2.0 containing raw data, we employ Vision Transformer (ViT) (Dosovitskiy et al., 2021) and Wav2Vec 2.0 (Baevski et al., 2020) to obtain the video and audio representations respectively. We note again that raw video and audio files are not available for **v1**, thus we develop the above architecture to uniformly handle both datasets, and leave how to best use multi-modal inputs in DeSIQ-2.0 for future work.

As social intelligence usually requires commonsense knowledge, we posit that injecting commonsense knowledge into the backbone language model in our architecture would improve the model's performance. Therefore, inspired by Jiang et al. (2021), we distill commonsense social knowledge from the following datasets into T5-small: Social Chemistry 101 (Forbes et al., 2020), ETHICS (Hendrycks et al., 2021) and Moral Stories (Emelin et al., 2021). Specifically, we pretrain

---

T5-small on these corpora and then finetune it on the downstream Social-IQ and DeSIQ datasets. We call this variant T5-small$_{Delphi}$.

### 4.1 Results on DeSIQ

We analyze the effectiveness of our proposed architecture, and the effect of the distillation of commonsense knowledge. The results of our new model architecture are shown in the bottom portions of Tables 4 and 5, where the inputs are concatenated ("✓" for the column "Concat"). We can make the following observations:

- Both T5-small and T5-small$_{Delphi}$ outperform the LSTM baseline on both $DeSIQ_d$ and $DeSIQ_s$ while not achieving near perfect accuracy, showing the effectiveness of our proposed architecture as well as the unbiased nature of DeSIQ.

- When the question is given as part of the model input, T5-small and T5-small$_{Delphi}$ (✓) significantly outperform the vanilla versions (✗), showing the effectiveness of our model architecture.

- Injecting commonsense knowledge can indeed improve model performance on social intelligence. T5-small$_{Delphi}$ with "q+a+t" inputs shows the best A2 score as $76.77\%$ and A4 and $74.51\%$ on $DeSIQ_d$, and $67.70\%$ in A2 on $DeSIQ_s$. On $DeSIQ_d$, it outperforms T5-small in all but one settings (q+a+v for A2). On $DeSIQ_s$, however, T5-small shows competitive performance, and significantly outperforms T5-small$_{Delphi}$ on A4 for both q+a+t. We leave the investigation of this result to future work.

- In many cases, adding the transcript can help improve model performance, and usually more effective than adding the video modality. Since T5-small$_{Delphi}$ is pretrained on a textual corpus, it is reasonable that adding the video modality may decrease model performance.

- Compared to $DeSIQ_d$, $DeSIQ_s$ is a more challenging dataset, as except for "a", performance of T5-small and T5-small$_{Delphi}$ drops for all others.

- Comparing the performance of q+a+t/q+a+v and q+a, we can observe that both T5-small and T5-small$_{Delphi}$ can learn some shortcuts, as they achieve comparable performance when only given the question and answers as input.

Some examples are shown in Appendix A Figure 5, illustrating the influence of different modalities. The first two examples show how the transcript and video features may provide clues for answering the

---

question. For instance, the first example cannot be correctly answered based on "q+a", since the transcript contains the required information. T5-small$_{Delphi}$ is the only model that predicts correct options for the last example in Figure 5, which we attribute to Delphi's commonsense knowledge.

## 4.2 Results on DeSIQ-2.0

For DeSIQ-2.0, we can apply multi-modal model using the raw videos and audios. The experimental results are in Table 6. Apart from some similar observations on DeSIQ-1.0 above, some new conclusions can be made as follows:

- Adding audios or videos can help improve model performance. Moreover, audios are more effective as the model achieves overall best A4 score of 74.13% under the "q+a+s" setting.

- Employing raw transcripts can reduce model performance ($57.23\%\to52.02\%$ ) as they are usually 5 times longer than other input features in length, which can largely influence the representation learning procedure of other inputs.

- Compared with ChatGPT in Table 7, our best result outperforms 24.52% on A4, which shows DeSIQ-2.0 to be a challenging dataset.

We also conduct experiments with settings "a+t" and "a+v", but don't include them in the paper. After debiasing, both settings for the proposed model on DeSIQ2.0 are near the random guess performance: "a+t" 22.66% and "a+v" 26.46%. Thus, questions are necessary when compared with the performance of "q+a+t" and "q+a+v" inputs in Table 6.

## 4.3 Further Research Questions

The above results show the lack of biases and challenging nature of our DeSIQ datasets as well as promising performance by modestly-sized language models. These results lead to the following important research directions for further investigation:

- Are there still noticeable biases in DeSIQ, and if so, how to further debias it?

- What is the performance of stronger language models on DeSIQ?

- How to effectively incorporate socio-cultural and commonsense knowledge into large language models for this task?

- How to utilize multi-modal language models to better exploit video and audio input?

## 5 Related Work

**Debiasing.** Shah et al. (2020) proposed a number of *expectations* to examine a **model**'s performance on a number of multiple-choice QA datasets and observed that the model (RoBERTa) falls short of the expectations. Different from this work, we establish a systematic methodology, consisting of six novel methods, to examine a **dataset**. And we design some experimental settings on both Social-IQ and Social-IQ-2.0.

Language Dependence/Prior is actually a **MODEL** side bias resulting in the model largely depending on one major modality (usually text). Reducing it can be regarded as an optimization problem. Gat et al. (2020) try to balance the influence of text and image from the MODEL side. Though the paper includes Social-IQ dataset and gets positive results, it doesn't realise the bias's existence in the original Social-IQ dataset.

Shortcut is a **DATA** side bias resulting in the model easily learning the pattern/repeated word in one dataset. For example, some keywords can occur both in the question and the correct answer, but not in the incorrect answers, so that the model directly gets clues from this overlap. Ye and Kovashka (2021) identify the shortcut and show its negative effects. However, they only modify the validation data and propose a masking approach to perform more robust training on the MODEL side.

In this paper, we start from the DATA side and also peform debiasing on the DATA side. Moreover, the bias we identify in the Social-IQ dataset is not the same kind, which is mainly in the answers and much harder to be debiased in the DATA side. Thus, though they share some similarities, we consider it a new task.

**Multi-modal Question Answering.** With different multiple input modalities, such as image and video, multi-modal question answering problem is more challenging and has been rising more and more attention in the past few years. Datasets like MovieQA (Tapaswi et al., 2016), TGIF-QA (Jang et al., 2017), TVQA (Lei et al., 2018) and TVQA+ (Lei et al., 2020) provide images, GIFs or video clips in addition to text-based single-turn questions. There are some datasets like AVSD (Alamri et al., 2019) that require dialogue history to predict answers for multi-turn questions. All these datasets evaluate model capacity of perceive the contextual information contained in both text and non-text modalities.

**Social Intelligence Learning.** Understanding and reasoning about social commonsense knowledge and human interactions is essential for cognitive social intelligence learning. Bosselut et al. (2019) present a comprehensive study on automatic commonsense knowledge base construction, which mines the intents and reasons behind human behaviors. (Jiang et al., 2021) propose a commonsense moral model to better understand social norms and make reliable ethical judgments on real-world human actions. In this paper, we focus on the Social-IQ dataset (Zadeh et al., 2019), a benchmark provides a diverse annotated set of videos and question-answer pairs. We run all the experiments on this dataset because it is much more related to social intelligence learning than other datasets.

## 6 Conclusion

Social intelligence is an essential ingredient for effective human-computer communications. In this paper, we analyze Social-IQ, a multiple-choice question answering benchmark dataset for social intelligence. Our empirical analysis reveal the severe biases present in Social-IQ, which can be easily exploited by modestly-sized language models such as T5-small to achieve perfect accuracy on its development set. We construct the DeSIQ benchmark by applying simple perturbation-based techniques on Social-IQ and show that the DeSIQ vastly reduce the biases in Social-IQ. Moreover, we propose a new model architecture on DeSIQ and set strong performance baselines for this challenging new dataset. Finally, our comprehensive analyses open up a number of important research questions for further investigation.

## Limitations

For the proposed model architecture designed to address the new DeSIQ benchmark, we mainly employ text-based language models and pretrain them on text-based corpora. The exploration of powerful multi-modal language models, instead of using the projection function as is done in this paper, is thus an important future research work direction. Due to resource constraints, all the experiments in this work were under conducted only once with the same random seed equals 42. Multiple runs with different random seeds would enable us to performance statistical significance tests of the results, and thus make the findings more reliable.

## Ethics Statement

Although the benchmark is designed for studying human behaviors and research purposes only, the resources and findings could be used unexpectedly. For example, it is possible that harmful content exists in the Social-IQ dataset, thus also in our DeSIQ datasets, based on which trainable models could turn from a positive to a negative perspective. Thus, it is prudent for researchers working on social intelligence to pledge to only make ethical use of our benchmark datasets.

## Acknowledgement

This material is based on research partially sponsored by the DARPA Assured Neuro Symbolic Learning and Reasoning (ANSR) program under award number FA8750-23-2-1016, the DARPA Knowledge Management at Scale and Speed (KMASS) program under award number HR00112220047, and the DARPA Computational Cultural Understanding (CCU) program under the agreement number HR001122C0029. The U.S. Government is authorised to reproduce and distribute reprints for Governmental purposes notwithstanding any copyright notation thereon. The authors are grateful to the anonymous reviewers for their helpful comments.

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

# A   Appendix

## A.1   Experimental Settings

For all experiments, we fix our random seed at 42. We run fully supervised learning on one A40 GPU with 40GB memory, and set the learning rate as 1e-4 as well as early stopping by monitoring the loss on the development set. Typically, it takes 2-3 hours using only features to finish 100-epoch training with a batch size of 8.

We also employ GPT-3 (175B parameters) as it is the current state-of-the-art language model in a variety of NLP tasks for in-context learning For the few-shot evaluation, we select top-3 most similar questions from the training set. We did not test other choice of k due to both budget and time constraints, which we leave for future work.

| Answer | Question | Transcript | Video | Prediction | | | | |
|---|---|---|---|---|---|---|---|---|
| | | | | | a | q+a | q+a+t | q+a+v |
| A (correct): He's upset the brunette man doesn't want his children raised with religion. B (incorrect): He's sad that he doesn't get along with the brunette man. | Why does the blonde man seem sad? | - Are your parents religious? - They are religious. … - And I think she's really softened my views on a lot of things. I just don't want my children to grow up with the same experience I had. |  | LSTM
T5-small
Delphi | B
B
B | B
B
B | A
A
A | B
A
B |
| `Transcript clues: yellow background words.` | | | | | | | | |
| A (incorrect): yes, he is being serious B (correct): no, he is being sarcastic for comedic affect | Is the man serious when he asks "are you captivated yet"? | - Are you captivated yet? So I'm gona to shuffle it |  | LSTM
T5-small
Delphi | A
A
A | A
A
A | A
A
A | A
B
B |
| `Video clues: another person are smiling.` | | | | | | | | |
| A (correct): It is not as high as she thought it would be B (incorrect): She is just acting upset to trick the opponent | Why is the woman upset when she receives her score? | - that's what it sounds like when they split from an evolutionary lineage - yeah you get 407 points - Oh |  | LSTM
T5-small
Delphi | B
B
B | B
B
B | B
B
A | B
B
A |
| `SocailChem101 clues (DELPHI pretraining corpus): it's fun to be happy about getting a high score.` | | | | | | | | |

Figure 5: Predictions of different models on DeSIQ$_d$ benchmark. We show the answers, question, transcript and video modalities in the first four columns, and predictions of different models using our settings of input features. Red ellipses are correct predictions based on certain features. Some explainable clues are shaded in yellow.