# OpenReview forum: "DeSIQ: Towards an Unbiased, Challenging Benchmark for Social Intelligence Understanding"
_EMNLP/2023/Conference — EMNLP 2023 Main_

### Official Review · Reviewer_BsAc · 2023-08-04

**Soundness:** 4

**Excitement:**

4: Strong: This paper deepens the understanding of some phenomenon or lowers the barriers to an existing research direction.

**Paper Topic And Main Contributions:**

This paper analyzes and reveals biases found In the Social-IQ dataset. The insights are used to generate a new dataset (DeSIQ) that reduces biases the biases seen in the initial dataset. This datasets are tested on different architectures such as T5-Small, GPT-3, etc.

Different configurations for the question-answer setting are tested such as: the replacement of incorrect answers, replacement of context, replacement of the correct answer, etc. The different model sizes are also tested to see how model-size affects the capability to learn/obtain a specific bias that does not generalize.

Experiments where hypothesis are replaces with other correct answers from the set (RIWA) shows large degradations in accuracy, and potential biases learned by these models.

The new DeSIQ dataset is also generated with different settings/contexts in mind to generate a more challenging set. Context from the same or different videos can be used to create different mixtures of the data.



**Questions For The Authors:**

The prompt used for the zero-shot experiments can be crafted in different ways. The context in the prompt could yield different results. Have you tried different prompts for these experiments (and maybe observed different results based on the context)?

**Reasons To Accept:**

This paper delivers a comprehensive analysis on different architectures and their biases when tested on DeSIQ data.
The new dataset also demonstrates how accuracy fluctuates when the same model is used, and how shortcuts can be learned when similar context is used for the incorrect answers.
This work is also well written and the experiments are sound. The ablation experiments also illustrates in different ways the biases found in the data.

**Reasons To Reject:**

Previous work (Removing Bias by Gat et al.) have focused on techniques to diminish the bias at the model level. The manipulation of the data has also been explored by Ye et al. ( Shortcut Effects in Visual Commonsense Reasoning). Although previous work explain techniques to manipulate the data/model, it does not focus on the specific DeSIQ dataset. Some overlapping concepts could be a weakness of the work

**Reproducibility:**

3: Could reproduce the results with some difficulty. The settings of parameters are underspecified or subjectively determined; the training/evaluation data are not widely available.

**Reviewer Confidence:**

3: Pretty sure, but there's a chance I missed something. Although I have a good feel for this area in general, I did not carefully check the paper's details, e.g., the math, experimental design, or novelty.

**Typos Grammar Style And Presentation Improvements:**

The tables on page 6 contain shaded results to illustrate relevant sections in the paper. While this is useful, it could also confuse readers who glance through the tables first.

---

> ### Author Rebuttal · Authors · 2023-08-26
>
> Q1: Previous work (Removing Bias by Gat et al.) have focused on techniques to diminish the bias at the model level. The manipulation of the data has also been explored by Ye et al. ( Shortcut Effects in Visual Commonsense Reasoning). Although previous work explains techniques to manipulate the data/model, it does not focus on the specific DeSIQ dataset. Some overlapping concepts could be a weakness of the work.
>
> A1: Thank you for pointing out these papers. Below we summarise the main differences of our work from these papers.
>
> Language Dependence/Prior is actually a MODEL side bias resulting in the model largely depending on one major modality (usually text). Reducing it can be regarded as an optimization problem. Gat et al. (Removing Bias in Multi-modal Classifiers) try to balance the influence of text and image from the MODEL side. Though the paper includes Social-IQ dataset and gets positive results, it doesn’t realise the bias's existence in the original Social-IQ dataset.
>
> Shortcut is a DATA side bias resulting in the model easily learning the pattern/repeated word in one dataset. For example, some keywords can occur both in the question and the correct answer, but not in the incorrect answers, so that the model directly gets clues from this overlap. Ye et al. (Shortcut Effects in Visual Commonsense Reasoning) identify the shortcut and show its negative effects. However, they only modify the validation data and propose a masking approach to perform more robust training on the MODEL side.
>
> In our paper, we start from the DATA side and also peform debiasing on the DATA side. Moreover, the bias we identify in the Social-IQ dataset is not the same kind, which is mainly in the answers and much harder to be debiased in the DATA side. Thus, though they share some similarities, we consider it a new task.
> We will include these discussions in our final version.
>
> Q2: The prompt used for the zero-shot experiments can be crafted in different ways. The context in the prompt could yield different results. Have you tried different prompts for these experiments (and maybe observed different results based on the context)?
>
> A2: Yes, the context design could largely influence the output quality. Unfortunately, we didn’t try other prompts and only used the prompts listed in our paper due to time and resource limitations. As zero-shot/few-shot learning is not the focus of this work,  we leave prompt optimization to future research work.
>
> Q3: The tables on page 6 contain shaded results to illustrate relevant sections in the paper. While this is useful, it could also confuse readers who glance through the tables first.
>
> A3: Thank you for mentioning that. We will explore other presentation options to improve their readability in our final version.

---

### Official Review · Reviewer_tCWT · 2023-08-05

**Soundness:** 4

**Excitement:**

4: Strong: This paper deepens the understanding of some phenomenon or lowers the barriers to an existing research direction.

**Paper Topic And Main Contributions:**

This paper presents a deep analysis of the Social-IQ dataset, a benchmark for studying social intelligence, as well as introduces a new dataset called DeSIQ. The authors identified substantial biases in the Social-IQ dataset which can be exploited by moderately strong language models to achieve perfect performance without even needing the context or the question.

The Social-IQ dataset was developed to evaluate social intelligence in AI models. It is a multiple-choice question answering (MCQA) dataset with questions, answer options, videos, and other multi-modal inputs. However, the study found that even relatively simple models could achieve 100% accuracy due to the biases present in the dataset. This led the authors to develop a new dataset, DeSIQ, using simple perturbations to the Social-IQ dataset to significantly reduce these biases.

**Questions For The Authors:**

The construction of unbiased, challenging datasets for social intelligence, like DeSIQ, requires considerable effort and a comprehensive understanding of the biases in existing datasets. What measures have the authors taken for that? Did they use any well-structured documents to construct guidelines?

**Reasons To Accept:**

By identifying and rectifying biases, new, more challenging datasets (like DeSIQ) can be constructed for better benchmarking of AI's social intelligence capabilities. The studies can shed light on the effect of different aspects of machine learning models (like model size, model style, learning settings, common-sense knowledge, and multi-modality) on performance.

**Reasons To Reject:**

The construction of unbiased, challenging datasets for social intelligence, like DeSIQ, requires considerable effort and a comprehensive understanding of the biases in existing datasets. What measures have the authors taken for that? Did they use any well-structured documents to construct guidelines?

**Reproducibility:**

3: Could reproduce the results with some difficulty. The settings of parameters are underspecified or subjectively determined; the training/evaluation data are not widely available.

**Reviewer Confidence:**

3: Pretty sure, but there's a chance I missed something. Although I have a good feel for this area in general, I did not carefully check the paper's details, e.g., the math, experimental design, or novelty.

---

> ### Author Rebuttal · Authors · 2023-08-26
>
> Q: The construction of unbiased, challenging datasets for social intelligence, like DeSIQ, requires considerable effort and a comprehensive understanding of the biases in existing datasets. What measures have the authors taken for that? Did they use any well-structured documents to construct guidelines?
>
> A: The identification of biases is based on our comprehensive empirical evaluation of the original dataset, on which a small model (i.e. T5-small) can achieve 100% accuracy with answer choices as the only input.
> For debiasing, we employed existing techniques in the literature, but found that the simple perturbation approach we proposed is the most effective. We will add more discussions on the debiasing approaches we experimented in the revision.

---

### Official Review · Reviewer_hZcY · 2023-08-05

**Soundness:** 3

**Excitement:**

4: Strong: This paper deepens the understanding of some phenomenon or lowers the barriers to an existing research direction.

**Paper Topic And Main Contributions:**

This paper defined a comprehensive methodology to identify the bias in Social-IQ, a dataset of multiple choice questions on videos of complex social interaction, and constructed a new unbiased and more challenging benchmark named DeSIQ. Experiments on both supervised learning and zero/few-shot learning were carried to prove the effectiveness of the Debiasing Approach. Besides, a new model architecture along with baselines on DeSIQ was proposed. In conclusion, the paper introduced the related work and discussed important research directions for further investigation.

**Questions For The Authors:**

1)	In the proposed benchmark DeSIQ, in order to debias the answers, the incorrect answers are substituted with correct answers of other questions. However, I wonder that whether the questions are still needed for models to achieve an excellent score, since the problem is transformed into a matching problem between answer and video/transcript,

2)	I also noticed that there were no experiment with the input of “a+t/v”.


**Reasons To Accept:**

1)	This paper defined a comprehensive methodology to identify bias in multiple choice question datasets. This is an interesting findings for the related study.

2)	The authors proposed to construct a unbiased challenging benchmark DeSIQ in the field of Social Intelligence Understanding.

3)	The authors conducted extensive experiments to prove its effectiveness, and proposed a new model architecture on DeSIQ along with baselines.


**Reasons To Reject:**

1)	DeSIQ’s data are totally from the existing dataset Social-IQ without expansion or furthur cleaning, which means that the qualities of those data are completely depend on Social-IQ.

2)	As mentioned in the paper, all experiments were conducted only once, which makes the results less convincing.


**Reproducibility:**

3: Could reproduce the results with some difficulty. The settings of parameters are underspecified or subjectively determined; the training/evaluation data are not widely available.

**Reviewer Confidence:**

3: Pretty sure, but there's a chance I missed something. Although I have a good feel for this area in general, I did not carefully check the paper's details, e.g., the math, experimental design, or novelty.

---

> ### Author Rebuttal · Authors · 2023-08-26
>
> Q1: DeSIQ’s data are totally from the existing dataset Social-IQ without expansion or further cleaning, which means that the qualities of those data are completely dependent on Social-IQ.
>
> A1: We’d like to emphasise that our aim is indeed to improve the quality of Social-IQ by debiasing it. While paraphrasing questions/answers could produce more training data, doing so may introduce other biases. Therefore, we will leave further cleaning and expansion to future research.
>
> Q2: As mentioned in the paper, all experiments were conducted only once, which makes the results less convincing.
>
> A2: Though we did our experiments only once, we chose the same random seed, ran 2-3 times for each setting, and got the same value on the accuracy score. If time and resources permit, we can run all the experiments with different random settings.
>
> Q3: In the proposed benchmark DeSIQ, in order to debias the answers, the incorrect answers are substituted with correct answers of other questions. However, I wonder whether the questions are still needed for models to achieve an excellent score, since the problem is transformed into a matching problem between answer and video/transcript.
>
> A3: The questions are indeed needed since the model should know which answer it needs to generate/match. Please see the answer of Q4 for a detailed explanation and empirical evidence.
>
> Q4: I also noticed that there was no experiment with the input of “a+t/v”.
>
> A4: We did conduct  experiments with settings “a+t” and “a+v”, but didn’t include them in the paper. After debiasing, both settings for the proposed model on DeSIQ2.0 are near the random guess performance: “a+t” 22.66% and “a+v” 26.46%. Thus, questions are necessary when compared with the performance of “q+a+t” and “q+a+v” inputs in Table 6. If applicable, we could involve these results in our final version.

---

### Meta-Review · Area_Chair_xdHj · 2023-09-19

**Recommendation:** 4

**Metareview:**

This paper describes a comprehensive method to analyze biases in Social-IQ dataset, and proposed a new dataset DeSIQ using a simple yet effective perturbation method from which biases are reduced.

Reviewers appreciated the importance of the problem space, the well-structured content, and solid experiments. They raised two concerns: 1) The lack of diversity in the data construction method: The perturbation method represents just one application of existing debiasing methods; and 2) The lack of generalizability of the proposed dataset: DeSIQ derives from only Social-IQ.

---

### Decision · Program_Chairs · 2023-10-07

**Decision:**

Accept-Main

**Comment:**

This paper describes a comprehensive method to analyze biases in Social-IQ dataset, and proposed a new dataset DeSIQ using a simple yet effective perturbation method from which biases are reduced.

Reviewers appreciated the importance of the problem space, the well-structured content, and solid experiments. They raised two concerns: 1) The lack of diversity in the data construction method: The perturbation method represents just one application of existing debiasing methods; and 2) The lack of generalizability of the proposed dataset: DeSIQ derives from only Social-IQ.